# Subcellular localization of type IV pili regulates bacterial multicellular development

Courtney K. Ellison [1,2,4] ✉, Chenyi Fei [1,2], Triana N. Dalia [3], Ned S. Wingreen [1,2], Ankur B. Dalia [3] ✉, Joshua W. Shaevitz [1] ✉ & Zemer Gitai [2] ✉

In mammals, subcellular protein localization of factors like planar cell polarity proteins is a key driver of the multicellular organization of tissues. Bacteria also form organized multicellular communities, but these patterns are largely thought to emerge from regulation of whole-cell processes like growth, motility, cell shape, and differentiation. Here we show that a unique intracellular patterning of appendages known as type IV pili (T4P) can drive multicellular development of complex bacterial communities. Specifically, dynamic T4P appendages localize in a line along the long axis of the cell in the bacterium *Acinetobacter baylyi*. This long-axis localization is regulated by a functionally divergent chemosensory Pil-Chp system, and an atypical T4P protein homologue (FimV) bridges Pil-Chp signaling and T4P positioning. We further demonstrate through modeling and empirical approaches that subcellular T4P localization controls how individual cells interact with one another, independently of T4P dynamics, with different patterns of localization giving rise to distinct multicellular architectures. Our results reveal how subcellular patterning of single cells regulates the development of multicellular bacterial communities.

Type IV pili (T4P) are thin, polymeric surface structures that are broadly distributed among bacteria and archaea and whose dynamic cycles of extension and retraction are essential to diverse processes including DNA uptake during natural transformation and biofilm formation[1–3]. T4P dynamics are achieved through the polymerization and depolymerization of pilin subunits by the activity of ATP-hydrolyzing motors. Imaging T4P and their dynamics in live cells was historically challenging due to their small size, but a recently-developed technique for fluorescently labeling pilin subunits has led to new discoveries and renewed interest in T4P physiology and function[4–6].

To date, T4P have only been reported to localize either to the cell poles or dispersed around the cell body, and no role for these differences in multicellular organization has been identified. We recently developed the gram-negative coccobacillus-shaped bacterium

*Acinetobacter baylyi* as a model for T4P study by introducing a cysteine residue into the major pilin ComP for subsequent labeling with fluorescent, thiol-reactive maleimide dyes[7]. *A. baylyi* is a soil-dwelling species that is closely related to emerging pathogens like *Acinetobacter baumannii*, and both use their T4P to take up DNA from the environment to acquire antibiotic resistance (AbR) genes[7,8]. Upon fluorescently labeling T4P in this species, we noticed that T4P are produced along a line that is parallel to the long axis of the cell body (Fig. 1a, Supplementary Fig. 1). In this study, we identify that this long-axis localization pattern is controlled by the Pil-Chp chemosensory system and show that FimV connects Pil-Chp signaling to T4P localization. We find that long-axis localization is important for the structural development of multicellular communities, highlighting how subcellular positioning of protein complexes can influence complex communities of microorganisms.

[1]Lewis-Sigler Institute for Integrative Genomics, Princeton University, Princeton, NJ, USA. [2]Department of Molecular Biology, Princeton University, Princeton, NJ, USA. [3]Department of Biology, Indiana University, Bloomington, IN, USA. [4]Present address: Department of Microbiology, University of Georgia, Athens, GA, USA. ✉e-mail: c.ellison@uga.edu; ankdalia@indiana.edu; shaevitz@princeton.edu; zgitai@princeton.edu

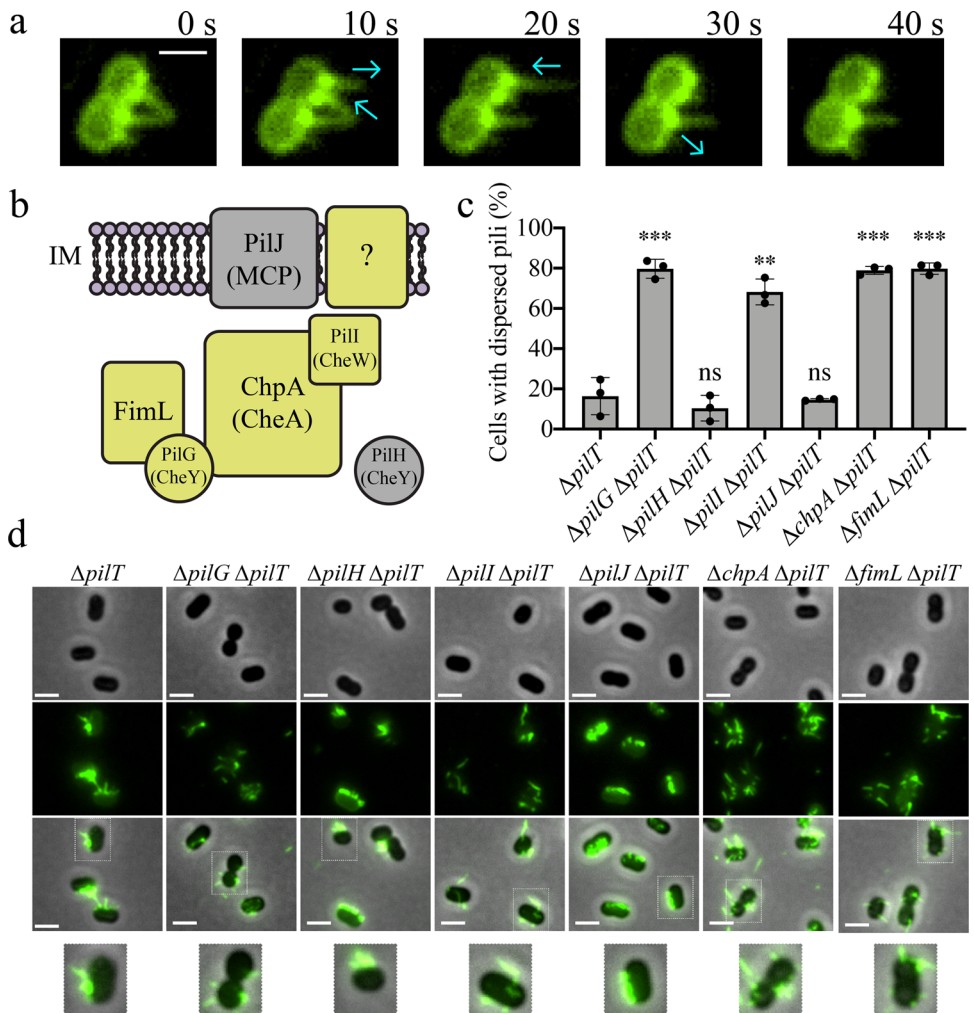

**Fig. 1 | *Acinetobacter baylyi* localizes its T4P in a line along the long axis of the cell dependent on the Pil-Chp pathway. a** Representative time-lapse images of *A. baylyi* extending and retracting fluorescently labeled T4P with background fluorescence subtracted. Blue arrows indicate direction of T4P movement. Scale bar, 1 μm. **b** Schematic of the Pil-Chp components found in *A. baylyi* with analogous flagellar chemotaxis protein names in parentheses. Deletions of components colored gold cause dispersed T4P localization while deletion of components in gray have no effect on localization of T4P. IM, inner membrane. **c** Quantification of the percentage of cells in a population with dispersed T4P. Cells with dispersed T4P were defined as cells that had T4P on multiple sides of the cell body. Each data point represents an independent, biological replicate and bar graphs indicate the

mean ± SD. For each biological replicate, a minimum of 70 total cells were assessed. Statistical comparisons were made using One-Way ANOVA followed by Dunnett's multiple comparisons test comparing log-transformed values from mutant strains to the Δ*pilT* parent: ns, not significant; **$P < 0.01$; ***$P < 0.001$. Exact measurements and $P$ values are reported in the Source Data file. **d** Representative images of *A. baylyi* Pil-Chp mutants displaying dispersed or linearly localized T4P with background fluorescence subtracted. Zoomed-in images of representative single cells from each strain are outlined in dashed boxes and shown below. Scale bars, 2 μm. Genotypes of each strain used in each figure panel are outlined in Supplementary Table 1.

## Results

### *Acinetobacter baylyi* localizes its T4P in a line along the long axis of the cell dependent on the Pil-Chp pathway

To identify factors involved in the unusual linear long-axis T4P localization of *A. baylyi*, we examined T4P in a collection of mutants in genes associated with T4P function in other species[7]. Many of these mutants displayed no T4P, inhibiting analysis of their effects on T4P localization, but one subset of mutants disrupted long-axis localization and instead resulted in dispersed T4P. These mutants belonged to the "Pil-Chp" chemotaxis-like system that integrates T4P-mediated surface sensing with T4P function and chemical signaling with twitching motility in *Pseudomonas aeruginosa*[9,10].

The Pil-Chp system shares homology to flagellar chemotaxis systems, and analogous components are inferred to carry out similar functions (Fig. 1b). In *P. aeruginosa*, the methyl-accepting chemotaxis protein (MCP) homologue PilJ is thought to signal to the CheA homologue ChpA, and the CheW homologues PilI and ChpC are

believed to act as intermediaries for Pil-Chp signaling leading to phosphoryl transfer from ChpA to the CheY homologues PilG and PilH. Other regulatory components encoded in the Pil-Chp operon include the putative transcriptional regulator ChpD as well as PilK and ChpB that are inferred to control the methylation state of PilJ.

Bioinformatic analysis revealed several differences in the Pil-Chp operons of *A. baylyi* and *P. aeruginosa* (Supplementary Fig. 2). Specifically, *A. baylyi* lacks several regulatory genes found in *P. aeruginosa* (*pilK*, *chpB*, *chpC*, *chpD*, and *chpE*). Both *P. aeruginosa* and *A. baylyi* also have a Pil-Chp-associated factor, *fimL*, which is ectopic to the main operon. To assess T4P localization, we generated deletions of each of the *A. baylyi* Pil-Chp genes and combined each deletion with a deletion in *pilT*, the retraction motor whose loss leads to hyperpiliation. Deletion of *pilI*, *chpA*, *pilG*, and *fimL* all abrogated T4P localization and resulted in dispersed T4P around the cell body (Fig. 1c, d). However, deletions of *pilH* and *pilJ* did not affect T4P patterning. The fact that long-axis T4P localization persisted in the absence of *pilJ* was

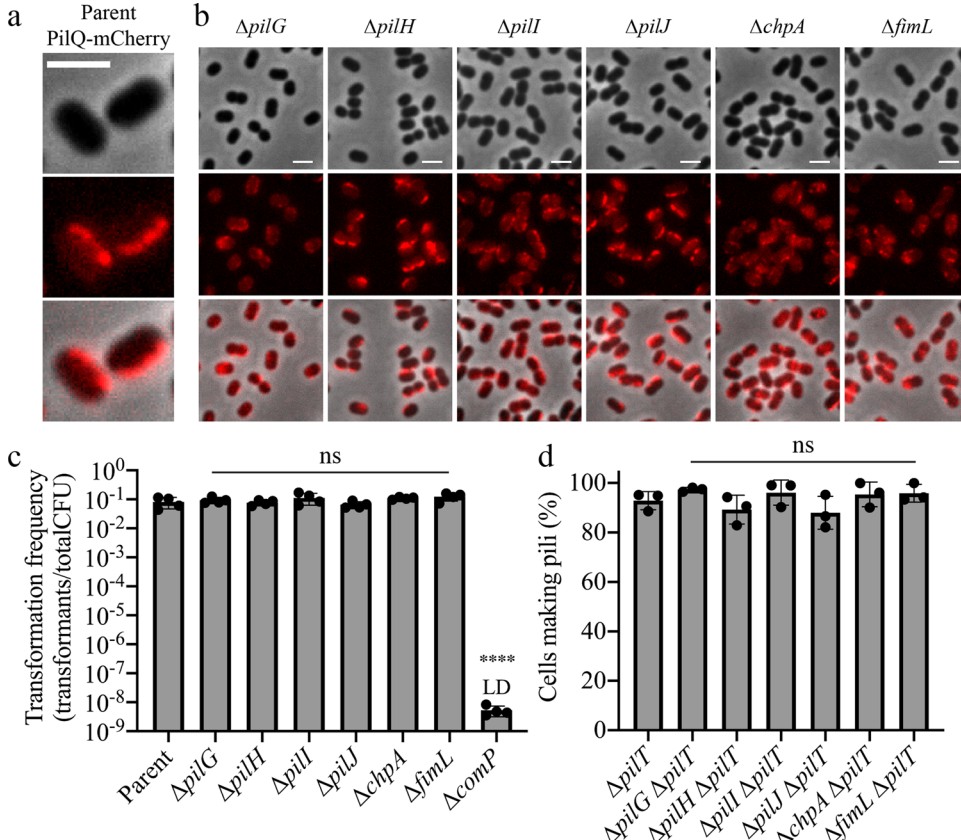

**Fig. 2 | The Pil-Chp pathway in *A. baylyi* has evolutionarily diverged in function to control T4P localization. a** Representative image of the parent strain containing a T4P machinery fluorescent mCherry fusion to the outer membrane secretin protein, PilQ, with background fluorescence subtracted. **b** Representative images of PilQ-mCherry localization in different Pil-Chp mutants with background fluorescence subtracted. **c** Natural transformation assays of indicated strains. Each data point represents a biological replicate and bar graphs indicate the mean ± SD. The transformation frequency of the Δ*comP* strain was below the limit of detection, indicated by LD. **d** Quantification of the percentage of cells in a population making

T4P based on the presence or absence of fluorescently labeled T4P. Each data point represents an independent, biological replicate and bar graphs indicate the mean ± SD. For each biological replicate, a minimum of 70 total cells were assessed. Scale bars, 2 μm. Statistical comparisons were made using One-Way ANOVA followed by Dunnett's multiple comparisons test comparing log-transformed values from mutant strains to the parent: ns, not significant; ****$P < 0.0001$. Exact measurements and $P$ values are reported in the Source Data file. Genotypes of each strain used in each figure panel are outlined in Supplementary Table 1.

particularly surprising because PilJ is the methyl-accepting chemotaxis receptor-like factor thought to act upstream of PilI and ChpA in *P. aeruginosa*[11]. These results suggest that there are fundamental differences in Pil-Chp function between *A. baylyi* and *P. aeruginosa* and that *A. baylyi* has a PilJ-independent Pil-Chp activity (Fig. 1b).

## The Pil-Chp pathway in *A. baylyi* has evolutionarily diverged in function to control T4P localization

We next tested whether the Pil-Chp pathway in *A. baylyi* regulates T4P synthesis as it does in other species including *Acinetobacter* pathogens[12]. In *P. aeruginosa*, T4P machinery is localized to both poles, but PilG controls which cell pole extends T4P presumably through interactions with the extension motor PilB[11,13]. We thus hypothesized that long-axis localization could either be due to long-axis localization of T4P machines themselves or the specific extension of T4P by a subset of dispersed machines through PilG activity along the long axis. PilQ is an outer-membrane-localized secretin that is a stationary T4P machinery component and therefore represents a good marker for T4P machinery[13]. An mCherry fusion to PilQ in *A. baylyi* (Supplementary Fig. 3) demonstrated that the T4P machinery exhibits the same long-axis localization pattern as labeled T4P (Fig. 2a). PilQ was also dispersed in Pil-Chp mutants that had dispersed T4P, suggesting that long-axis T4P result from localization of the machinery rather than

from regulating T4P synthesis (Fig. 2b). To more directly test whether T4P synthesis is regulated by Pil-Chp components, we performed natural transformation assays as prior work established that T4P are essential for DNA uptake in *A. baylyi*[7,14]. Pil-Chp mutants exhibited equal levels of natural transformation compared to the parent (Fig. 2c). Furthermore, direct examination of labeled T4P revealed similar production of T4P in Pil-Chp mutants and the parent strain (Fig. 2d). Together these results demonstrate that the Pil-Chp pathway is an adaptable system that has evolved in *A. baylyi* to regulate T4P machinery localization instead of T4P synthesis as in other systems.

## The Pil-Chp pathway controls T4P localization through the positioning of a divergent FimV homologue

To dissect the molecular mechanisms underpinning T4P patterning via Pil-Chp we initially focused on FimL. While our results indicate that the function of the Pil-Chp proteins is different in *A. baylyi* and *P. aeruginosa*, the presence of core Pil-Chp proteins in both systems suggests that the molecular interactions between Pil-Chp pathway components may be conserved. In *P. aeruginosa*, FimL interacts with the T4P assembly protein FimV to recruit PilG to T4P machinery[15,16]. Unable to detect a FimV homologue using bioinformatic tools, we instead applied a targeted proteomics approach to identify FimL-binding proteins that may link the Pil-Chp pathway to T4P to control their

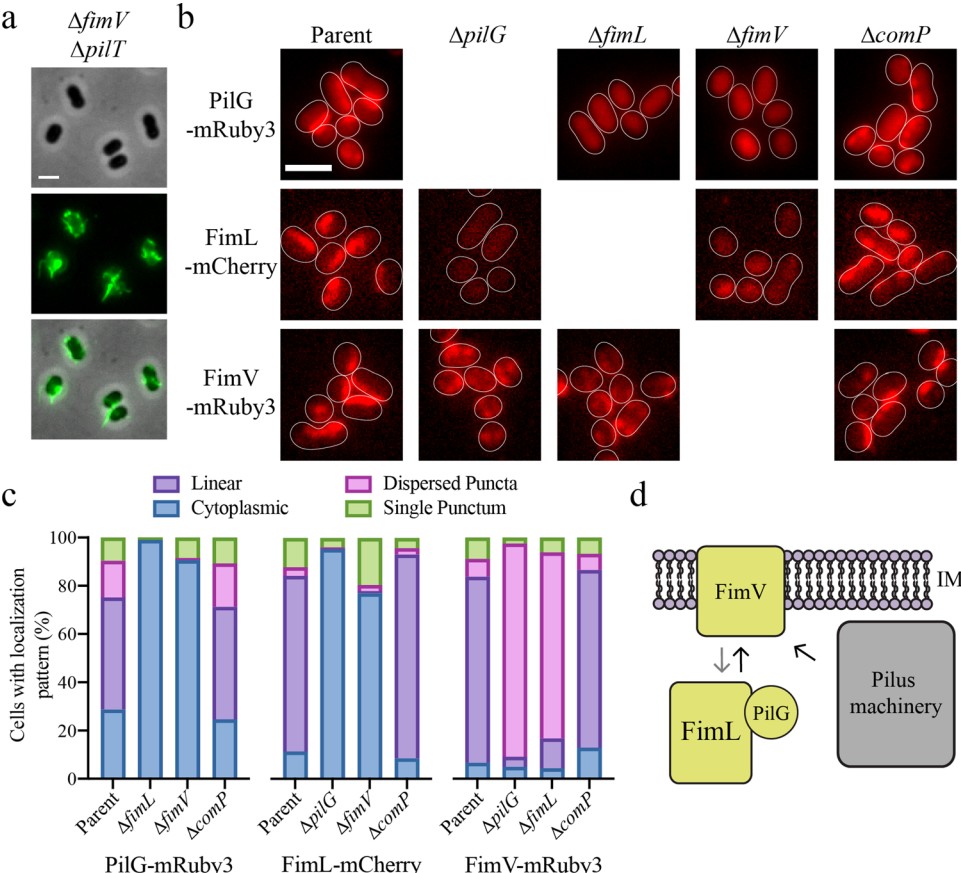

**Fig. 3 | The Pil-Chp pathway controls T4P localization through the positioning of a divergent FimV homologue. a** Representative image of a Δ*fimV* Δ*pilT* mutant displaying dispersed T4P labeled with AF488-maleimide dye and with background fluorescence subtracted. **b** Representative images of indicated protein fusion strains with background fluorescence subtracted. White outlines are cell body outlines from phase contrast images. **c** Quantification of fluorescence phenotypes depicted in **b**. Data are compiled from two independent biological replicates, and a minimum of 100 total cells were assessed for each strain. **d** Schematic of the proposed molecular pathways connecting the Pil-Chp system to T4P localization. Gray arrow indicates the dependence of FimV linear localization on the Pil-Chp pathway while black arrows indicate the recruitment of FimL/PilG and T4P machinery proteins to FimV. Deletions of components colored gold cause dispersed T4P localization while deletion of components in gray have no effect on localization of T4P machines. IM, inner membrane. Scale bars, 2 μm. Genotypes of each strain used in each figure panel are outlined in Supplementary Table 1.

localization. We made a FimL-3xFLAG construct and performed coimmunoprecipitation experiments using α-FLAG magnetic beads followed by mass spectrometry to identify potential interacting partners. Using this approach, the top identified protein was encoded by gene ACIAD0477. A BLAST search for conserved domain homology identified a predicted inner membrane protein with similarity to the C-terminus of *P. aeruginosa* FimV (Supplementary Fig. 4)[17]. Deletion of ACIAD0477, which we hereafter refer to as *fimV*, resulted in dispersed T4P localization, suggesting that this factor plays a role in regulating T4P positioning through the Pil-Chp system (Fig. 3a).

We assessed the spatial regulation of T4P patterning by Pil-Chp signaling by fluorescently tagging the newly-identified FimV homologue and FimL. We also fluorescently tagged PilG, since it is predicted to act downstream of PilI and ChpA function[15]. All three proteins localized similarly with fluorescently labeled T4P in a line along the long axis of the cell, although we note that the PilG-mRuby3 fusion was only partially functional (Supplementary Fig. 5). Deletions of *fimV* and *fimL* resulted in diffuse, cytoplasmic localization of PilG with occasional puncta (Fig. 3b, c, Supplementary Fig. 6), and FimL-mCherry fluorescence in Δ*pilG* and Δ*fimV* strains was likewise cytoplasmically diffuse (Fig. 3b, c). In contrast, *fimL* and *pilG* deletions resulted in dispersed puncta of FimV (Fig. 3b, c). To address the possibility that the phenotypes resulted from reduced FimL levels rather than

localization, we ectopically expressed *fimL* and found that it restored T4P localization in a Δ*fimL* mutant but did not restore FimV or PilG localization in either Δ*pilG* or Δ*fimV* backgrounds (Supplementary Fig. 6). FimV is the only Pil-Chp component other than PilJ that is predicted to be a membrane protein. Thus, PilG and FimL are required for long axis localization of FimV, while FimV likely recruits FimL and PilG to the cell membrane where T4P are assembled (Fig. 3d). Deletion of T4P did not affect localization of PilG, FimL, or FimV, highlighting that *A. baylyi* Pil-Chp regulation is independent of T4P themselves or T4P-mediated activities like DNA uptake or surface sensing. Together these data support a model where Pil-Chp signaling controls FimV positioning to regulate T4P localization (Fig. 3d).

## T4P localization controls multicellular interactions in *A. baylyi*

Having established the molecular connection between the Pil-Chp system and subcellular T4P localization, we next sought to determine the biological consequences for dispersed vs. linear long-axis T4P. *A. baylyi* lacks genes encoding a flagellum and can grow as either individual cells or in a multicellular biofilm-like community[18]. T4P are known to regulate biofilm formation in other bacteria so we considered the possibility that T4P might affect *A. baylyi* multicellular biofilm formation[19,20]. In some species, including *P. aeruginosa*, twitching motility is thought to play a critical role in biofilm

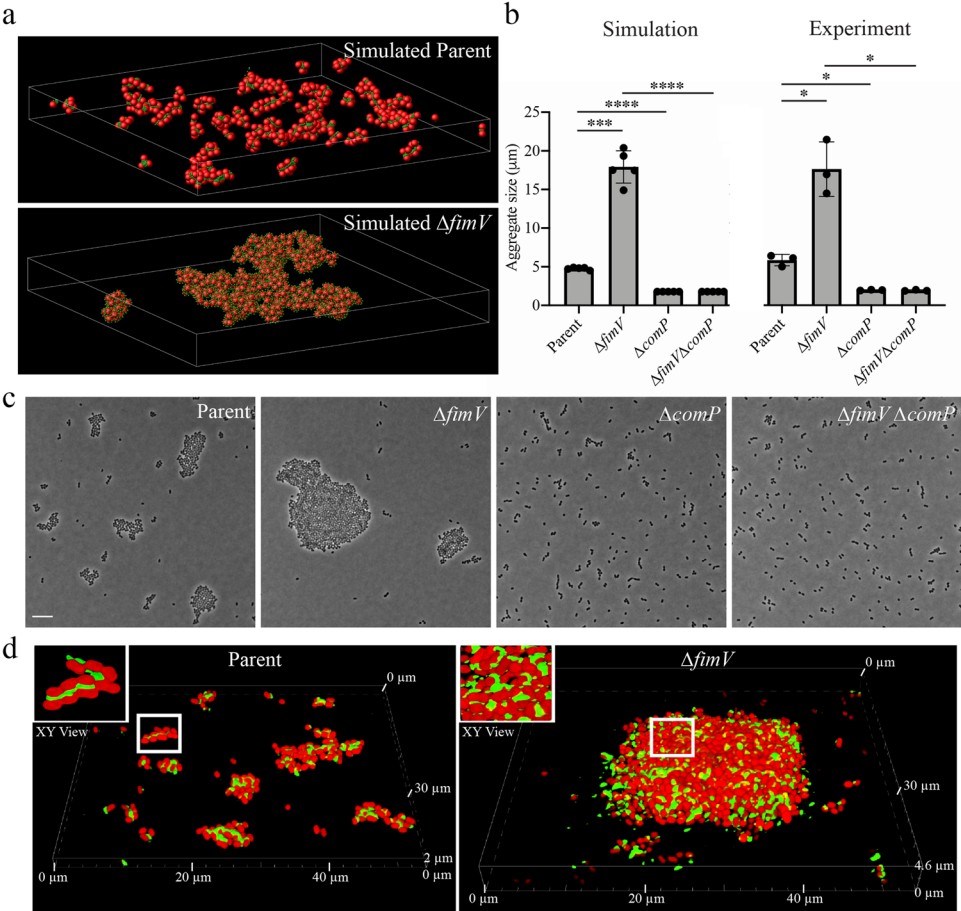

**Fig. 4 | T4P localization controls multicellular interactions in *A. baylyi*.**
**a** Representative images of simulated data for modeled cellular objects with different patterns of adhesin localization. **b** Quantification of multicellular aggregate sizes from simulated or experimental data collected for indicated strains. Each data point represents a simulation ($n = 5$) or biological ($n = 3$) replicate and bar graphs indicate the mean ± SD. Statistical comparisons were made using two-tailed Welch's *t*-test: *$P < 0.05$; ***$P < 0.001$; ****$P < 0.0001$. Exact measurements and $P$ values are reported in the Source Data file. **c** Representative images of multicellular aggregates from experimental data for indicated strains in a Δ*pilT* background. Scale bar, 2 μm. **d** Representative confocal microscopy images collected for indicated strains. Green areas are labeled T4P and red areas are cell bodies expressing cytoplasmic mRuby3. Insets show a magnified top-down view of cells in the white square on each panel. Genotypes of each strain used in each figure panel are outlined in Supplementary Table 1.

development[1]. However, *A. baylyi* does not exhibit classical twitching motility under laboratory growth conditions (Supplementary Fig. 7). To explore general principles of how the geometric distribution of T4P might affect multicellular interactions in nonmotile cells, we first used a computational approach. Specifically, we performed molecular dynamics simulations comparing interactions between "cellular objects" possessing self-adhesive structures ("adhesins") that were either localized to one side of the object or dispersed around each object (similar to T4P in wildtype or *fimV* mutants, respectively (Supplementary Fig. 8)). Simulation data show that asymmetric localization of adhesins results in small groups of cells, whereas dispersed adhesin localization results in larger multicellular aggregates (Fig. 4a, b, Supplementary Fig. 9). This finding suggests that localizing adhesins together sterically reduces the ability of any one cell to interact with multiple cells, thereby limiting multicellular connectivity and large-scale aggregation.

To experimentally test our simulation's prediction, we examined multicellular aggregate formation in *A. baylyi*. We leveraged our discovery of the dispersed T4P in Δ*fimV* to compare multicellular development in cells with patterned or dispersed nanomachines. We also performed our experiments with non-dynamic Δ*pilT* strains to avoid complications that could be introduced by variability in T4P dynamics. Our experimental data closely mirrored our simulations, with dispersed T4P mutants producing much larger aggregates of cells than

the long-axis T4P parent (Fig. 4b, c). We confirmed that this difference was due to T4P, as deletion of the major pilin gene *comP* in both Δ*fimV* and parent strains disrupted cell aggregation (Fig. 4b, c). This result also highlights the critical role that T4P play in maintaining cell-cell contact. Deletions of *fimV* or *comP* had no effect on growth rates (Supplementary Fig. 10), reinforcing the conclusion that cell-cell aggregation is dependent on T4P localization and not growth.

Our simulations and experimental data comparing Δ*fimV* and parent strains suggested T4P localization regulates the organization of multicellular structures through self-associations maintained at the junctions of interacting cells (Fig. 4a, b, c). To further validate this model, we sought to observe whether fluorescently-labeled T4P are indeed found between individual cells in multicellular aggregates. Unfortunately, protocols for fluorescently labeling T4P include several wash steps that often disrupt cell-cell contact (Fig. 1d). We thus developed a protocol to first label T4P in single cells and then allow cells to establish cell-cell contact through additional incubation under normal growth conditions. Confocal imaging experiments confirmed that cells with labeled T4P can form multicellular structures (Fig. 4d). As predicted, the T4P in the parent cells localized at cell-cell junctions, demonstrating that T4P sustain intercellular connections. Meanwhile, Δ*fimV* multicellular aggregates exhibited T4P dispersed between multiple cells in much larger three-dimensional structures. While multicellular structures made by the parent strain are typically a single

layer of cells in height, Δ*fimV* multicellular aggregates appear as multiple layers of stacked cells interconnected by T4P (Fig. 4d).

## Discussion

In this work, we identified a subcellular pattern of T4P localization that *A. baylyi* uses to control intercellular interactions and regulate the organization of multicellular communities. We show that subcellular localization of T4P in *A. baylyi* is dependent on a truncated homologue of the pilus-associated protein FimV found in *P. aeruginosa* in which it regulates T4P assembly[16]. In *P. aeruginosa*, FimV is a large 919 amino acid protein that is predicted to localize to the inner membrane with the C-terminus residing in the cytoplasm and the N-terminus resting in the periplasm. The periplasmic domain possesses a peptidoglycan-binding motif that is predicted to bind peptidoglycan and aid assembly of the outer membrane secretin of the T4P assembly complex[16]. In contrast, the C-terminal cytoplasmic domain is predicted to be involved in intracellular signaling. In *A. baylyi*, FimV lacks the predicted periplasmic domain found in *P. aeruginosa*, in-line with our results that the Pil-Chp pathway in *A. baylyi* does not regulate T4P synthesis. Instead, it is possible that truncation of FimV in *A. baylyi* influenced its evolution into a localization factor. While FimV is clearly a localization determinant of T4P machinery in *A. baylyi*, the details of FimV localization remain unclear. It is possible that filament-forming proteins including cytoskeletal elements may act as a scaffold for FimV localization, or it may be that that Pil-Chp signaling allows FimV to form a filament that localizes parallel to the long axis of the cell.

We show that deletion of the MCP homologue PilJ does not affect T4P localization. Canonical MCP proteins rest in the inner membrane and act as key signaling receptors during chemotaxis. Because PilJ does not play a role in Pil-Chp mediated T4P localization and no alternative MCP homologues are encoded in the genome, we hypothesize that an alternative, unidentified signaling protein fills this role (Fig. 1b). Because PilJ is conserved in *A. baylyi* while other Pil-Chp genes have been lost (Supplementary Fig. 2), it is likely that PilJ acts as a signaling receptor for alternative Pil-Chp functions yet to be discovered.

While T4P dynamics are important for biofilm formation in many species, the factors that dictate T4P dynamics remain poorly understood. We thus used mutants lacking T4P dynamics in our studies to focus our efforts on understanding the function of T4P localization in *A. baylyi* biofilm development. Our findings demonstrate that T4P localization controls the spatial organization of multicellular communities by regulating the location and number of cell-cell interactions that occur. Biofilms with cells capable of T4P dynamics may reveal more interesting changes in multicellular architecture over time. Because cells use their T4P to stick to each other, dynamic T4P may provide an avenue for cells to reorient themselves in multicellular communities over longer temporal scales and in response to different environmental conditions. With this new understanding of how T4P localization influences the structure of microbial communities, future work can focus on identifying the factors involved in regulating T4P dynamics and the role of those dynamics in biofilm formation. In many species including *P. aeruginosa*, which harbors polar T4P that are essential for biofilm formation, large three-dimensional biofilms are commonly observed. Our data show that T4P localization can influence multicellular architecture, and it is thus possible that the polar localization of T4P in *P. aeruginosa* plays a crucial role in its biofilm formation.

At the molecular level we reveal that the localization of T4P to the long axis is mediated by an adapted form of a conserved chemosensory pathway. At the functional level our findings show how subcellular organization can pattern multicellular architecture. Specifically, we find that similar to how subcellular localization of planar cell polarity proteins can lead to tissue-scale patterning in mammals, subcellular localization of T4P nanomachines can lead to multicellular-scale patterning in bacteria. These data thus extend our understanding of how intracellular molecular patterning can evolve to influence biology at both microscopic and macroscopic scales.

## Methods

### Bacterial strains and culture conditions

*Acinetobacter baylyi* strain ADP1 was used throughout this study. For a list of strains used throughout, see Table S1. *A. baylyi* cultures were grown at 30 °C in Miller lysogeny broth (LB) medium unless otherwise indicated and on agar supplemented with kanamycin (50 μg/mL), spectinomycin (60 μg/mL), gentamycin (30 μg/mL), chloramphenicol (30 μg/mL), and/or apramycin (30 μg/mL) where appropriate.

### Construction of mutant strains

Mutants in *A. baylyi* were made using natural transformation as described previously[7,21]. Mutant constructs were made by splicing-by-overlap (SOE) PCR to stich (1) ~3 kb of the homologous region upstream of the gene of interest, (2) the mutation where appropriate (for deletion by allelic replacement with an AbR cassette, or the fusion protein), and (3) ~3 kb of the homologous downstream region. For a list of primers used to generate mutants in this study, see Table S2. The upstream region was amplified using F1 + R1 primers, and the downstream region was amplified using F2 + R2 primers. All AbR cassettes were amplified with primers ABD123 (ATTCCGGGGATCCGTCGAC) and ABD124 (TGTAGGCTGGAGCTGCTTC). Fusion proteins were amplified using the primers indicated in Table S2. In-frame deletions were constructed using F1 + R1 primer pairs to amplify the upstream region and F2 + R2 primer pairs to amplify the downstream region with ~20 bp homology to the remaining region of the downstream region built into the R1 primer and ~20 bp homology to the upstream region built into the F2 primer. To construct the *fimL-3×FLAG* strain, the sequence encoding the 3×FLAG tag was built into the R1 and F2 primers. SOE PCR reactions were performed using a mixture of the upstream and downstream regions, and middle region where appropriate using F1 + R2 primers. SOE PCR products were added with 50 μl of overnight-grown *A. baylyi* cultures to 450 μl of LB in 2 ml round-bottom microcentrifuge tubes (USA Scientific) and grown at 30 °C rotating on a roller drum for 3–5 h. For Ab^R-constructs, transformants were serially diluted and plated on LB and LB + antibiotic. For unmarked in-frame deletions and protein fusion constructs, cells were serially diluted and plated without selection on LB. In-frame deletions were confirmed by PCR using primers ~150 bp up and downstream of the introduced mutation, and fusions were confirmed by sequencing.

The PilQ-mCherry fusion was linked to a gentamycin resistance cassette (Supplementary Fig. 3) to permit easier transformation through selection on gentamycin. Because *pilQ* is in an operon upstream of the essential gene *aroK*, we placed *aroK* under a $P_{tac}$ constitutive promoter to prevent deleterious polar effects on its expression. The upstream region containing the C-terminal end of PilQ was amplified using primers CE590 + CE196, mCherry was amplified using CE197 + CE198, the gent cassette regions were amplified using CE637 + CE593 and CE594 + CE595, the $P_{tac}$ promoter was amplified using primers CE596 + CE336, and the downstream regions were amplified using primers CE638 + CE592. SOE PCRs and natural transformation were then carried out as described above.

The *fimL* complementation strain was constructed as described previously[7] by placing the gene of interest under a constitutive $P_{tac}$ promoter at the *vanAB* locus. First, the *vanAB* genes were replaced with a kanR cassette as described previously[7]. Then, a $P_{tac}$ promoter was introduced using primers CE317 + CE260 to amplify the upstream region, CE261 + CE336 to amplify the $P_{tac}$ promoter, CE947 + CE948 to amplify *fimL*, and CE406 + CE176 to amplify the downstream region. SOE PCR and natural transformation were then carried out as described above.

The *$P_{tac}$-mRuby3* constitutive expression strain was constructed similarly, with a few exceptions. First, Ptac-mRuby3 was introduced into

the *kan* cassette using primers CE317 + CE260 to amplify the upstream region, CE261 + CE262 to amplify *P_{tac}-mRuby3*, and CE263 + CE176 to amplify the downstream region. SOE PCR and natural transformation were then carried out as described above to obtain a strain carrying Δ*vanAB::kan*, *P_{tac}-mRuby3*. To use the constitutive *P_{tac}-mRuby3* construct in strains already containing *kan* resistance cassettes, we replaced the kanamycin resistance gene with an apramycin resistance gene. Using the Δ*vanAB::kan*, *P_{tac}-mRuby3* strain as a template, we amplified the upstream arm using primers CE317 + CE1511 and the downstream arm using primers CE1512 + CE176. Next, we used primers CE1509 + CE1510 to amplify the apramycin resistance gene. SOE PCRs and natural transformation were then carried out as described above.

### Natural transformation assays

Assays were performed as previously described[7,22]. Strains were grown overnight in LB broth at 30 °C on a roller drum. Then, 50 μl overnight cultures were subcultured into fresh LB medium and 50 ng of tDNA was added. In this study, a ΔACIAD1551::Spec^R PCR product (with 3 kb arms of homology on either side) was used as the tDNA. Reactions were incubated with end-over-end rotation at 30 °C for 5 h and then plated for quantitative culture on selective antibiotic plates (to quantify transformants) and on plain LB plates (to quantify total viable counts). Data are reported as the transformation frequency, which is defined as the (CFU/mL of transformants)/(CFU/mL of total viable counts).

### Pilin labeling, epifluorescence microscopy, and quantification

Pilin labeling was performed as described previously[4–7]. For pilus labeling in *A. baylyi*, 100 μl of overnight-grown cultures was added to 900 μl of LB in 1.5 ml microcentrifuge tube, and cells were grown at 30 °C rotating on a roller drum for 70 min. Cells were then centrifuged at 18,000 × g for 1 min and then resuspended in 50 μl of LB before labeling with 25 μg/ml of AlexaFluor488 C_5-maleimide (AF488-mal) (Invitrogen) for 15 min at room temperature. Labeled cells were centrifuged, washed once with 100 μl of LB without disrupting the pellet, and resuspended in 5–20 μl LB. Cells were imaged on a Nikon TiE microscope using a Plan Apo 100X objective, a Hamamatsu ORCA-Flash4.0 camera, and Nikon NIS Elements Imaging Software (Version 4.6). Cell bodies were imaged using phase contrast microscopy while labeled pili were imaged using fluorescence with a GFP filter set. To visualize T4P dynamics, cells were grown and treated exactly as described above except with M63 medium (1 mM MgSO_4, 0.5% casamino acids, 0.4% glucose, pH 7.0) instead of LB. Fluorescently tagged proteins were imaged in overnight cultures using fluorescence with an mCherry filter set. Cell numbers, the percent of cells making pili, the percent of cells making dispersed pili, and fluorescent protein fusion phenotypes were quantified manually using the cell counter plugin in ImageJ[23]. All imaging was performed under 1% UltraPure agarose (Invitrogen) pads made with PBS.

### Coimmunoprecipitation (pulldown) experiments

Coimmunoprecipitation experiments were performed as described previously[7], with some differences. Cultures of cells were grown shaking overnight in 30 ml LB in 125 ml volume flasks at 30 °C. The total culture volume was then harvested at 10,000 × g for 10 min at room temperature, and the supernatant was removed. Cell pellets were resuspended in 2 ml of Buffer 1 (50 mM Tris-Cl pH 7.4, 150 mM NaCl, 1 mM EDTA) and transferred to 2 ml volume microcentrifuge tubes and centrifuged at 18,000 × g for 1 min. Cells were washed once more with 2 ml of Buffer 1, and washed pellets were resuspended in 1 ml of Buffer 2 (50 mM Tris-Cl pH 7.4, 150 mM NaCl, 1 mM EDTA, 10 mM MgCl_2, 0.1% Triton X-100, 2% glycerol). To lyse cells, 4200 units of Ready-Lyse lysozyme (Lucigen), 30 units of DNase I (New England Biolabs), and 10 μl of concentrated protease inhibitor cocktail (Sigma) (one pellet dissolved in 500 μl of Buffer 1) were added to cell suspensions and incubated at room temperature for 45 min to 1 h. Cell debris

was removed by centrifugation at 10,000 × g for 5 min at 4 °C. 50 μl aliquots of α-FLAG magnetic bead slurry (Sigma) in 1.5 ml microcentrifuge tubes were washed three times with 1 ml of Buffer 2 using a magnetic collection stand. 1 ml of cell lysates was added to washed magnetic beads and subjected to end-over-end rotation at 4 °C for 2 h. Beads were then washed three times with 0.5 ml of Buffer 2, with 10 min incubations in Buffer 2 at 4 °C between each wash step. Beads were briefly washed a 4th time with 0.5 ml Buffer 2. To elute proteins from α-FLAG beads, 100 μl of elution buffer (150 μg/ml 3X-FLAG peptide, Sigma, in Buffer 2) was added and samples were subjected to end-over-end rotation at 4 °C for 30 min. Eluates were mixed with 4X SDS running buffer (40% v/v glycerol, 250 mM Tris-HCl pH 6.8, 0.8% w/v bromophenol blue, 20% v/v β-mercaptoethanol, 8% w/v SDS) and heated to 100 °C for 10–15 min before separation on a 4–20% pre-cast SDS-PAGE gel (BioRad) for -15 min so that the total protein was contained in the top -1 cm of the gel. Using a new razor blade between each sample, the -1 cm part of the gel containing each sample was removed and placed in a 1.5 ml microcentrifuge tube. Gel slices were then subjected to mass spectrometry analysis as described below.

### Mass spectrometry

In-gel digestion of protein bands using trypsin was performed as in ref. [24]. Trypsin digested samples were dried completely in a SpeedVac and resuspended with 20 μl of 0.1% formic acid pH 3 in water. 2 μl (-360 ng) was injected per run using an Easy-nLC 1200 UPLC system. Samples were loaded directly onto a 45 cm long 75 μm inner diameter nano capillary column packed with 1.9 μm C18-AQ resin (Dr. Maisch, Germany) mated to metal emitter in-line with an Orbitrap Fusion Lumos (Thermo Scientific, USA). Column temperature was set at 45 °C and 2 h gradient method with 300 nl per minute flow was used. The mass spectrometer was operated in data dependent mode with the 120,000 resolution MS1 scan (positive mode, profile data type, AGC gain of 4e5, maximum injection time of 54 s and mass range of 375–1500 m/z) in the Orbitrap followed by HCD fragmentation in ion-trap with 35% collision energy. Dynamic exclusion list was invoked to exclude previously sequenced peptides for 60 s and maximum cycle time of 3 s was used. Peptides were isolated for fragmentation using quadrupole (1.2 m/z isolation window). Ion-trap was operated in Rapid mode.

Raw files were searched using Byonic[25] and Sequest HT algorithms[26] within the Proteome Discoverer 2.2 suite (Thermo Scientific, USA). 10 ppm MS1 and 0.4 Da MS2 mass tolerances were specified. Carbamidomethylation of cysteine was used as fixed modification, oxidation of methionine, deamidation of asparagine and glutamine were specified as dynamic modifications. Pyro glutamate conversion from glutamic acid and glutamine are set as dynamic modifications at peptide N-terminus. Acetylation was specified as dynamic modification at protein N-terminus. Trypsin digestion with maximum of two missed cleavages were allowed. Files were searched against UP000000430 *Acinetobacter baylyi* database downloaded from Uniprot.org.

Scaffold (version Scaffold_4.11.1, Proteome Software Inc., Portland, OR) was used to validate MS/MS based peptide and protein identifications. Peptide identifications were accepted if they could be established at >95.0% probability by the Scaffold Local FDR algorithm. Protein identifications were accepted if they could be established at >99.9% probability and contained at least 2 identified peptides. Protein probabilities were assigned by the Protein Prophet algorithm[27].

### Twitching assays

Strains were streaked for single colony isolation onto LB agar plates and grown at 37 °C for *P. aeruginosa* or 30 °C for *A. baylyi*. Using a toothpick, a single colony was picked from each plate and stabbed into a fresh plate containing LB 1.5% agar LB until the toothpick touched the bottom of the plate to ensure bacteria were introduced at the plate-agar interface. Plates were then incubated in humid chambers at 37 °C for *P. aeruginosa* or 30 °C for *A. baylyi* for 4 days. Afterwards, the agar

was removed from the plates and 0.1% (w/v) crystal violet was used to stain cells on the bottom of the petri plates. Excess crystal violet was removed by a brief rinse with water before imaging.

### Imaging and quantification of *A. baylyi* multicellular aggregates

Strains were grown overnight in LB broth at 30 °C on a roller drum. Then, 1 μl of overnight cultures were subcultured into 3 ml of fresh LB medium and grown for an additional 16 h before imaging under a 1.5% agar LB pad. For two-dimensional image acquisition, aggregates were imaged using phase contrast on a Nikon TiE microscope using a Plan Apo 100X objective, a Hamamatsu ORCAFlash4.0 camera, and Nikon NIS Elements Imaging Software.

For three-dimensional image acquisition and labeling of T4P in biofilms, cells were labeled for T4P after 16 h growth as described above before resuspension in 1 ml of LB medium in a 1.5 ml microcentrifuge tube followed by end-over-end rotation at 30 °C on a roller drum for an additional 2–3 h. Aggregates were then imaged at 100 nm Z-plane intervals on a Nikon Ti2 microscope coupled with a CSU-W1 SoRa spinning disk using a 488 nm laser for imaging pili and a 594 nm laser for imaging fluorescent cell bodies, an Apo TIRF DIC 60X objective, a Hamamatsu ORCA-BT-Fusion camera, and Nikon NIS Elements Imaging Software. Images were denoised and visualized using standard settings and volume viewer in Nikon NIS Analysis software.

Because most cells were aggregated together in *fimV* mutants, some fields of view had very few cells present, and for this reason we only imaged fields with >50 cells present. We noticed that all *A. baylyi* strains used in this study have subpopulations of cells that form "rafts" of single cells with regular spacing between them in what appears to be an extracellular matrix. Because this behavior was T4P-independent, we avoided these groups of cells during imaging.

We developed a custom python-based protocol to process phase-contrast images of experimental cells and quantify the typical size of cell aggregates. Prior to quantification, images were binarized into cell regions ($\phi = 1$) and void regions ($\phi = 0$) using adaptive thresholding. Standard morphological filters, including opening and closing, were subsequently applied to the binary images to remove salt-and-pepper noise. We measured the normalized spatial autocorrelation functions of $\phi$, i.e., $C_{\phi\phi}(r) = \frac{\langle\phi(0)\phi(r)\rangle}{\langle\phi^2\rangle}$, where the average is taken over all pairs of pixels. We fitted an exponential decay with an offset, $C_{\phi\phi}(r) = C_\infty + (1 - C_\infty)\exp(-r/a)$, to the measured $C_{\phi\phi}(r)$ to obtain the correlation length $a$. We reported $2a$ as the size of cell aggregates. For the simulation datasets, we projected the cells at the end of the simulation onto the $x$–$y$ plane to generate the binary images, to which we applied the same protocol above to quantify the sizes of simulated cell aggregates.

### Molecular dynamics simulation

Molecular dynamics simulations are performed using LAMMPS (4 Feb 2020 version)[28]. We use two types of "atoms" to model the cell body (B) and the pilus (P), respectively. The radius of the P-atom sets the unit length, i.e., $\sigma_P = 1$, and the radius of the B-atom is set to be $\sigma_B = 5$. Cells are represented by rigid-body molecules with 1 centered B-atom and 19 surrounding P-atoms. The P-atoms are closely packed into a hexagonal aggregate for the simulated parent strains, and they are uniformly distributed around the B-atom for the simulated $\Delta fimV$ strains (Supplementary Fig. 8).

Simulations are done in the NVE ensemble using a Langevin thermostat. The interaction energies are normalized so that $k_B T = 1$. Specifically, interactions between two B-atoms and between a B-atom and a P-atom are described by repulsive truncated and shifted Lennard-Jones potentials, $E_{XY}(r) = \epsilon + 4\epsilon\left[\left(\frac{\sigma_X + \sigma_Y}{r}\right)^{12} - \left(\frac{\sigma_X + \sigma_Y}{r}\right)^6\right]$ with $(X,Y) = (B, B)$ or $(B, D)$, $\epsilon = 1$ and cutoff at $r = 2^{1/6}(\sigma_X + \sigma_Y)$. Binding between two P-atom occurs via a soft potential,

$E_{PP}(r) = A\left[1 + \cos(\pi r/r_c)\right]$ for $r < r_c$, with cutoff $r_c = \sigma_P$. We set the binding strengths to be $A = -8$ and $A = 0$ for the simulated *comP*$^+$ and *comP*$^-$ strains, respectively.

We set the average time it takes for a single P-atom to diffuse a unit length to be the unit of time $\tau = 1$. The simulation time step is 0.05 $\tau$. We simulate 512 cells in a $500 \times 500 \times 500$ box with periodic conditions. We run 5 replicates for each simulated strain. The system is first simulated for $8 \times 10^8 \tau$ to approach equilibrium, and we record 300 system configurations, one every $0.5 \times 10^6 \tau$ in the final $1.5 \times 10^8 \tau$ period (Supplementary Fig. 9). To mimic the experimental assays, each configuration is then compressed along the *z*-axis over $2 \times 10^7 \tau$ prior to aggregate-size quantification. The uniaxial compression is performed by two infinitely large parallel plates, which are initially separated by a distance of 500 and which move toward each other with a constant speed until reaching a final separation of 50.

### Growth curves

Strains were grown overnight in LB broth at 30 °C on a roller drum. Then, 1 μl of overnight cultures were subcultured into 150 μl fresh LB medium and grown in a 96-well plate in a Tecan Infinite M200 Pro microplate reader at 30 °C shaking for 16 h. Optical density (OD$_{600}$) was measured every 10 min.

### Statistics and reproducibility

All statistical tests were performed using GraphPad Prism 8 (Version 8.4.3). All microscopy images are representative of data from at least three independent, biological replicates.

### Reporting summary

Further information on research design is available in the Nature Research Reporting Summary linked to this article.

## Data availability

Source data are provided with this paper. Where indicated in methods, we searched the UP000000430 *Acinetobacter baylyi* database downloaded from Uniprot.org. Source data are provided with this paper.

## Code availability

All simulation code and data, and custom code for image processing and quantification are available on GitHub[29]: https://github.com/f-chenyi/pilus-localization; https://doi.org/10.5281/zenodo.7072658.

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

## Acknowledgements

We would like to thank the Proteomics and Mass Spectrometry Core Facility at Princeton University and S. Kyin for help with mass spectrometry. We would also like to thank the Confocal Microscopy Facility at Princeton University and G. Laevsky for help with confocal imaging. C.K.E. is a Damon Runyon Fellow supported by the Damon Runyon Cancer Research Foundation (DRG-2385-20). This work was supported in part by the National Science Foundation, through the Center for the Physics of Biological Function (PHY-1734030). This work was supported by National Institutes of Health grants R35GM128674 awarded to A.B.D. and R01 GM082938 awarded to N.S.W., and by National Institutes of Health Pioneer Award 1DP1AI124669-01 awarded to Z.G.

## Author contributions

C.K.E. and T.N.D. performed the experiments. C.F. and N.S.W. designed and C.F. performed simulation experiments. C.K.E., C.F., T.N.D., N.S.W., A.B.D., J.W.S., and Z.G. analyzed and interpreted data. C.K.E. and Z.G. wrote the paper with input from C.F., T.N.D., N.S.W., A.B.D., and J.W.S.

## Competing interests

The authors declare no competing interests.
