## [Peer Review File · Nature Communications]

Subcellular localization of type IV pili regulates bacterial multicellular developmentReviewer #1 (Remarks to the Author):

The manuscript by Ellison et al. describes work wherein fluorescent labeling of type IV pili (T4P) and pilus-related molecules is employed to characterize the spatial organization of this machinery in *Acinetobacter baylyi*. The authors show that pili localize along the long axis of the elongated cells, which contrasts the usual polar localization in other cells. A functionally divergent Pil-Chp system is shown to play a role for this localization. Combining coarse-grained stochastic simulations and experiments, the authors show that the distinct pilus localization affects the structure and size of bacterial colonies. The experimental results, mirroring the simulation results, demonstrate that cells with rather homogeneously distributed T4P produce much larger cell aggregates than the wild-type cells where T4P are laterally distributed along a line.

The manuscript stands out from other biological work due to the combination of advanced labeling techniques with more standard microbiological approaches and simulations. The data analysis and the simulations are solid (see more below) and the methodology is overall very sound and presented in adequate detail. Most noteworthy is a conceptual aspect, namely that local arrangement of microscopic adhesion machinery determines the mesoscopic structure of multicellular aggregates. This exemplary insight, as well as the advanced molecular-biological techniques, will make reading the article useful for other researchers, specialists and non-specialists. In the reviewer's opinion, this work is clearly one of the 10% best articles that appeared in this field during recent years. It is recommended to publish the article after revision.

Minor comments:

A few typos can be corrected throughout the text

L46: "Pil-Chp chemotaxis-like system that integrates T4P-mediated surface sensing with T4P function". It would be good to mention twitching chemotaxis (phosphor-lipids/sugar) for which the Pil-Chp system is thought to be at least as important as for surface sensing.

L51: it is noteworthy that the methylesterase ChpB and the methyltransferase PilK are missing, suggesting that the Pil-Chp system is not designed for adaptation here and likely not working as a surface-sensing receptor at all. Have the authors tested the hypothesis that that some other signaling pathway is taking over the role of lipid/sugar chemoreceptors in the context of twitching?

L53: The authors' finding that the Histidine Kinase ChpA deletion abrogates T4P localization, while PilJ is unessential is quite unintuitive and conflicts with the insinuation of Fig 1b. It would be helpful if this finding is somehow carried over into the figure.

L63: "T4P machinery is localized to both poles, but PilG controls which cell pole extends T4P"... this is a point that is not completely clear. In *P. aeruginosa*, PilG affects directional reversal, but that may be simply due to its direct effect on the polymerization ATPase PilB. What is the role of the PilB homologue in the present system? Does PilG interact with it?

L103: The essential, unanswered question is of course, how the sketched molecular interactions between FimV at the membrane and PilG/FimL results in the line-like arrangement of the Pili. While it is comprehensible that the authors prefer to address this in a follow up, readers would benefit from a short discussion of possible mechanisms.

L117: The simulations are essentially used to obtain statistics for aggregates built of objects with specific shapes and binding site distributions. To obtain appropriate results,

the details of the simulations probably do not matter since only the role of geometry is studied. Nevertheless, it would be desirable to show a proof that the systems are equilibrated and to show size distributions, rather than only the mean values.

L117: The simulation results only match the experimental data qualitatively. More quantitative results could probably be obtained if the authors vary the distribution, size, and shape of the small and large beads to capture the true geometry.

Reviewer #2 (Remarks to the Author):

Courtney Ellison and colleagues investigate the subcellular localization of type 4 pili in *Acinetobacter baylyi*. Using their previously established method for pilus fluorescence labeling, they show that pili localize along one side of the bacteria. Generating deletion strains of components of the che/chp system, they find that a subset of this system is required for one-sided localization in a pilT deletion background. They identify a membrane-associated component required for pilus localization and term it FimV. Taken together, they show that FimL, PilG, and FimV govern the localization of the type 4 pilus complexes, independent of the pilus machinery itself. They also show that the molecular mechanism controlling pilus localization is distinct from the mechanism that controls pilus generation in *Pseudomonas aeruginosa*, where the pilus complexes are static in contrast to *A. baylyi*. Finally, the authors address the effect of one-sided pilus location on the development of colonies. Using simulations and experiments, they show that in a non-motile background, the colonies formed by bacteria with one-sided localization are smaller than the colonies with peritrichous piliation.

In my opinion, the results are very interesting and timely. In particular, localization of type 4 pili along a line of the major axis of the cell body, is quite unexpected and has interesting implications for pilus-mediated functions. Also, by contrast to other type 4 pilus generating organisms like *P. aeruginosa* and *M. xanthus*, the pilus core complex is not stable. I find the major part of this manuscript (the mechanistic part) convincing. The manuscript is very clearly written. However, I have a major criticism concerning the last part dealing with the function of one-sided pilus localization. These results are not very profound and therefore, the title "regulates bacterial multicellular development" is an overstatement in my opinion. I suggest either toning this down and introducing an extended discussion of this point or conducting additional experiments that address the colony architecture.

Major point:

The authors claim in the title that "Subcellular localization of nanomachines regulates bacterial multicellular development". I was a bit disappointed to see that all they show is that colonies formed by a DpilV DpilT strain are somewhat larger on average than DpilT colonies. To me, this is a logic consequence of having pili only at one side of the cell body. The fact that the distribution of pili in colonies is different if the subcellular localization is different is reassuring, but also expected.

Furthermore, I am concerned about the use of a DpilT mutant background. It has been shown in many studies that pilus dynamics strongly affect the "development" of bacterial colonies, their architecture, and fluidity (in particular for *Neisseria* species). If the authors claim that pilus localization is functionally important for *A. baylyi*, then they should study colonies in the wt background with dynamic pili. If they don't want to study this aspect, I suggest that they explain why and add a discussion of this point.

Other points:

Line 45: It would be helpful to explain the functions of the genes involved in the Pil-Chp system in the Introduction.

Line 56: The authors are surprised by the absence of an effect of pilJ deletion. Do they

speculate that the localization does not respond to external stimuli? It would be good to discuss this point.

Line 110: The authors claim that *A. baylyi* is a nonmotile species. To my knowledge they are capable of twitching motility.

Line 125: If the authors decide not to investigate the effect of pilus dynamics, then they should mention this in the abstract.

Line 143: The discussion is overall very short. I suggest discussing the effect of subcellular type 4 pilus localization on multicellular interactions/colony development in comparison to other species with polar pili and peritrichous pili. Also, as mentioned before, the (putative) effects of pilus dynamics could be discussed at this point.

Reviewer #3 (Remarks to the Author):

This work, titled "Subcellular localization of nanomachines regulates bacterial multicellular development" shows how the positioning of Type IV Pilus (Tfp) motors in a line along the cell major axis, is a driver of *Acinobacter Baylyi* bacterial cell organization within biofilms. The Pil-Chp chemosensory-like system is responsible for this particular Tfp subcellular localization by the aid of a protein, FimV, that connects Pil-Chp with Tfp. FimV is not part of the Pil operon as in *P. aeruginosa*. It is instead an orphan gene whose encoded product is here identified as a FimL interactant by pull-down experiments using FimL as a bait. Finally, a computational approach shows that this pili localization along the cell main axis is responsible of medium-size cell group formation, whereas a peritrichous distribution of pili would lead to the formation of larger groups. This work is novel and original and definitely adds insights on the current knowledge of Tfp biology and functions.

I believe that some additional data and information could improve the manuscript and clarify some aspects. Here are some proposed additions/changes.

MAJOR COMMENTS

The text can be highly improved by the additional of some information and clarifications.
Line 13: What are planar cell polarity proteins? Please briefly explain and add a reference.

Line 19: It took me a while to correlate the sentence "tfp appendages localize in a line along the long cell axis" with what I saw on Figure 1a. I first thought that the authors meant that the filaments (appendages) ran parallel to the long axis which is not what Fig 1a shows. This figure shows indeed that a maleimide stained duplicating cell (or a diplococcus?) have four foci running parallel to the main cell axis. Filaments extruding from these foci are perpendicular to this axis. Please, better describe this localization pattern in the abstract and throughout the text, specifying that motors (rather than the appendages) localize in a line along the cell axis. An illustration showing a typical *Acinobacter baylyi* cell with the long cell axis indicated could also be added to Figure 1a.

Line 38: Why did the authors choose to use *Acinobacter baylyi* as a model to study Tfp? What is known on the biology of this bacterium? What is its cell shape? Is it a pathogen? What is the current knowledge on their Tfp? Are these bacteria motile when they are not in a biofilm? If tfp have a role in adhesion beside in transformation, do they adhere to each other or to some cellular matrix?

Figure 1d and 2b: It is useless to show 3 or 4 small cells when we have the histogram for statistical purposes. It is better to show one representative cell per strain where details are visible, especially for strains where pili are "dispersed". Do pili from Δ pilT strains localize in one line (one dimension) along the long cell axis or, because cells of this strains are hyper-piliated, pili occupy a larger surface as compared to wt? Would electron microscopy unveil these differences?

Line 54: I understand the use of Δ pilT to better visualize pili. However, authors should show that pil-chp mutations have similar effects in wildtype as in Δ pilT. It does not matter if differences are less pronounced in Δ pilT than in wt and if wt data are shown as

supplementary information. On the same line, in Figure 4c and 4d, what is the effect of a pilT deletion on biofilm formation as compared to wildtype? Pilus dynamics might be disturbing for the computational analysis, yet they are physiological and must play an important role in biofilm formation. So, are differences between biofilms of pil-chp mutants and the parental strain still relevant in a wildtype background as in Δ pilT?
Line 56: Deletions in pilJ have no effects on pili localization. Are other MCPs in *Acinobacter baylyi* that could signal to the CheA-like? Otherwise, what is the hypothesized function for PilJ? Please discuss.
Line 70: Are strains shown in 2b also in a Δ pilT background? If yes, please, specify.
Lines 81-90. The pull-down is a huge experiment that allowed to uncover the *A. baylyi* homolog of FimV, which is a very important result. This experiment deserves more emphasis maybe by adding a supplementary table with the hits.

MINOR COMMENTS

Line 16: Beside "growth and motility", add "cell shape and differentiation".

Line 36: Please use the term peritrichous instead than dispersed here and throughout the text.

Line 51: Please specify either in the text or in the figure legend the correspondent flagellar chemotaxis homologs.

Legend to Figure 3a: specify that pili are maleimide labelled.

Line 88: Any hypothesis on why FimV is truncated in *A. baylyi*? What is the fimV genetic environment?

Line 93: "localize similarly" instead than "colocalize".

Extendend Data4: quantification of dispersed pili and transformation efficiency are more appropriate ways to assess protein functionality.

Figure 3d: Is the arrow going from pilus machinery to FimV inverted?

Line 232: what the need of Fiji if protein fusion phenotypes have been quantified manually?

Line 288: "1 ul" is irrelevant without the dilution factor.

Line 302: >50 cells

All figure legends should contain strain names and genotypes.

A schematic representation of the hypothetical protein organization of *Acinobacter baylyi* pili would be helpful.

REVIEWER COMMENTS

Reviewer #1 (Remarks to the Author):

The manuscript by Ellison et al. describes work wherein fluorescent labeling of type IV pili (T4P) and pilus-related molecules is employed to characterize the spatial organization of this machinery in *Acinetobacter baylyi*. The authors show that pili localize along the long axis of the elongated cells, which contrasts the usual polar localization in other cells. A functionally divergent Pil-Chp system is shown to play a role for this localization. Combining coarse-grained stochastic simulations and experiments, the authors show that the distinct pilus localization affects the structure and size of bacterial colonies. The experimental results, mirroring the simulation results, demonstrate that cells with rather homogeneously distributed T4P produce much larger cell aggregates than the wild-type cells where T4P are laterally distributed along a line.

The manuscript stands out from other biological work due to the combination of advanced labeling techniques with more standard microbiological approaches and simulations. The data analysis and the simulations are solid (see more below) and the methodology is overall very sound and presented in adequate detail.

Most noteworthy is a conceptual aspect, namely that local arrangement of microscopic adhesion machinery determines the mesoscopic structure of multicellular aggregates. This exemplary insight, as well as the advanced molecular-biological techniques, will make reading the article useful for other researchers, specialists and non-specialists. In the reviewer's opinion, this work is clearly one of the 10% best articles that appeared in this field during recent years. It is recommended to publish the article after revision.

We would like to thank the reviewer for their very positive and helpful comments on our manuscript. We have addressed their comments and feel this has added clarity and value to the story. See below for details.

Minor comments:

A few typos can be corrected throughout the text

L46: "Pil-Chp chemotaxis-like system that integrates T4P-mediated surface sensing with T4P function". It would be good to mention twitching chemotaxis (phosphor-lipids/sugar) for which the Pil-Chp system is thought to be at least as important as for surface sensing.

We have added this information to the text: "the "Pil-Chp" chemotaxis-like system that integrates T4P-mediated surface sensing with T4P function and chemical signaling with twitching motility in *Pseudomonas aeruginosa*" (lines 51-52).

L51: it is noteworthy that the methylesterase ChpB and the methyltransferase PilK are missing, suggesting that the Pil-Chp system is not designed for adaptation here and likely not working as a surface-sensing receptor at all. Have the authors tested the hypothesis that that some other

signaling pathway is taking over the role of lipid/sugar chemoreceptors in the context of twitching?

Under our lab conditions, *A. baylyi* does not exhibit twitching motility so we cannot test this hypothesis. *Acinetobacter* species exhibit a surface-spreading phenotype that is commonly confused with twitching motility, but this spreading has been shown to be pilus-independent (Harding et al. MBio 2013). We have added data demonstrating a lack of twitching by classical twitch assays in these growth conditions to the extended data. This does not rule out that there is some condition where twitching may occur and where the Pil-Chp pathway may play a role, but it does highlight that for our experimental conditions twitching motility does not likely play a role in the phenotypes we report. We have changed the text to incorporate these results:

“In some species, including *P. aeruginosa*, twitching motility is thought to play a critical role in biofilm development. However, *A. baylyi* does not exhibit classical twitching motility under laboratory growth conditions (Extended data 7).” (lines 124-127).

L53: The authors’ finding that the Histidine Kinase ChpA deletion abrogates T4P localization, while PilJ is unessential is quite unintuitive and conflicts with the insinuation of Fig 1b. It would be helpful if this finding is somehow carried over into the figure.

We have altered fig 1b to hopefully clarify this point, and we have added discussion related to the finding that ChpA is required for localization while PilJ is not. Specifically, we hypothesize that an additional membrane protein could be signaling through ChpA, and that PilJ likely has an unidentified function. We have added a “mystery” protein to the figure including the text:

“We show that deletion of the MCP homologue PilJ does not affect T4P localization. Canonical MCP proteins rest in the inner membrane and act as key signaling receptors during chemotaxis. Because PilJ does not play a role in Pil-Chp mediated T4P localization and no alternative MCP homologues are encoded in the genome, we hypothesize that an alternative, unidentified signaling protein fills this role (Figure 1b). Because PilJ is conserved in *A. baylyi* while other Pil-Chp genes have been lost (Extended Data 2), it is likely that PilJ acts as a signaling receptor for alternative Pil-Chp functions yet to be discovered.” (lines 177-183).

L63: “T4P machinery is localized to both poles, but PilG controls which cell pole extends T4P”... this is a point that is not completely clear. In *P. aeruginosa*, PilG affects directional reversal, but that may be simply due to its direct effect on the polymerization ATPase PilB. What is the role of the PilB homologue in the present system? Does PilG interact with it?

We agree with the reviewer that PilG likely affects directional reversal through its effect on PilB, and this was a point we failed to communicate. We have clarified this in the text: “PilG controls which cell pole extends T4P presumably through interactions with the extension motor PilB” (lines 74-75).

Additionally, the reviewer asks about the role of the PilB protein in *A. baylyi* in the present system. In our last paper (Ellison et al. Nature Communications 2021), we identified that in addition to PilB, a second extension motor homologue TfpB is also critical for T4P extension. In that paper, we made clean deletions of both extension motor genes, and neither played a role in the localization phenotype.

L103: The essential, unanswered question is of course, how the sketched molecular interactions between FimV at the membrane and PilG/FimL results in the line-like arrangement of the Pili. While it is comprehensible that the authors prefer to address this in a follow up, readers would benefit from a short discussion of possible mechanisms.

We have added discussion of possible mechanisms of FimV localization.

“While FimV is clearly a localization determinant of T4P machinery in *A. baylyi*, the details of FimV localization remain unclear. It is possible that filament-forming proteins including cytoskeletal elements may act as a scaffold for FimV localization, or it may be that that Pil-Chp signaling allows FimV to form a filament that localizes parallel to the long axis of the cell.” (lines 173-176).

L117: The simulations are essentially used to obtain statistics for aggregates built of objects with specific shapes and binding site distributions. To obtain appropriate results, the details of the simulations probably do not matter since only the role of geometry is studied. Nevertheless, it would be desirable to show a proof that the systems are equilibrated and to show size distributions, rather than only the mean values.

We thank the reviewer for raising this important point. To check whether the simulation systems had reached equilibrium, we monitored the change in the total energy of the system over time (Extended Data 9). Our previous simulations were stopped at $t = 108$ simulation time units, at which point the simulated parent and $\Delta fimV$ systems had not fully reached equilibrium.

To ensure equilibration of the system, we performed new simulations lasting for 8×10^8 simulation time units. As requested, we added a new figure panel (Extended Data 9a) to show that the total energy of the system had levelled off by the end of the simulation, indicating that the system had reached equilibrium. We also included a new figure panel (Extended Data 9b) showing the size distributions of the simulated aggregates. We have updated Fig. 4 with new simulation results, and revised the Methods section to reflect the change in the simulation protocol:

“The system is first simulated for $8 \times 10^8 \tau$ to approach equilibrium, and we record 300 system configurations, one every $0.5 \times 10^6 \tau$ in the final $1.5 \times 10^8 \tau$ period (Extended Data 9). To mimic the experimental assays, each configuration is then compressed along the z-axis over $2 \times 10^7 \tau$ prior to aggregate-size quantification” (lines 393-396).

L117: The simulation results only match the experimental data qualitatively. More quantitative

results could probably be obtained if the authors vary the distribution, size, and shape of the small and large beads to capture the true geometry.

We thank the reviewer for this suggestion. As shown in the revised Fig. 4b, the new results of the equilibrated simulation systems match quantitatively with the experimental data.

Reviewer #2 (Remarks to the Author):

Courtney Ellison and colleagues investigate the subcellular localization of type 4 pili in *Acinetobacter baylyi*. Using their previously established method for pilus fluorescence labeling, they show that pili localize along one side of the bacteria. Generating deletion strains of components of the che/chp system, they find that a subset of this system is required for one-sided localization in a pilT deletion background. They identify a membrane-associated component required for pilus localization and term it FimV. Taken together, they show that FimL, PilG, and FimV govern the localization of the type 4 pilus complexes, independent of the pilus machinery itself. They also show that the molecular mechanism controlling pilus localization is distinct from the mechanism that controls pilus generation in *Pseudomonas aeruginosa*, where the pilus complexes are static in contrast to *A. baylyi*. Finally, the authors address the effect of one-sided pilus location on the development of colonies. Using simulations and experiments, they show that in a non-motile background, the colonies formed by bacteria with one-sided localization are smaller than the colonies with peritrichous piliation.

In my opinion, the results are very interesting and timely. In particular, localization of type 4 pili along a line of the major axis of the cell body, is quite unexpected and has interesting implications for pilus-mediated functions. Also, by contrast to other type 4 pilus generating organisms like *P. aeruginosa* and *M. xanthus*, the pilus core complex is not stable. I find the major part of this manuscript (the mechanistic part) convincing. The manuscript is very clearly written.

We thank the reviewer for their comments and have incorporated a number of text changes to address their concerns. We feel this has deepened the discussion and clarified several points in the manuscript. See below for details:

However, I have a major criticism concerning the last part dealing with the function of one-sided pilus localization. These results are not very profound and therefore, the title "regulates bacterial multicellular development" is an overstatement in my opinion. I suggest either toning this down and introducing an extended discussion of this point or conducting additional experiments that address the colony architecture.

We disagree with the reviewer that our results are not very profound. First, it was unknown that the T4P of *A. baylyi* played a role in intercellular adhesion, a finding which has not been demonstrated in many other biofilm-forming species including *P. aeruginosa*.

Second, our data reveal that the heights of $\Delta fimV$ communities are very different from those of the parent strain. We have added text to point out this difference in height which can be seen in figure 4d by noting the Z-axis scale bars:

“While multicellular structures made by the parent strain are typically a single layer of cells in height, $\Delta fimV$ multicellular aggregates appear as multiple layers of stacked cells interconnected by T4P (Fig. 4d).” (lines 157-159).

Major point:

The authors claim in the title that "Subcellular localization of nanomachines regulates bacterial multicellular development". I was a bit disappointed to see that all they show is that colonies formed by a DpilV DpilT strain are somewhat larger on average than DpilT colonies. To me, this is a logic consequence of having pili only at one side of the cell body. The fact that the distribution of pili in colonies is different if the subcellular localization is different is reassuring, but also expected.

Furthermore, I am concerned about the use of a DpilT mutant background. It has been shown in many studies that pilus dynamics strongly affect the "development" of bacterial colonies, their architecture, and fluidity (in particular for *Neisseria* species). If the authors claim that pilus localization is functionally important for *A. baylyi*, then they should study colonies in the wt background with dynamic pili. If they don't want to study this aspect, I suggest that they explain why and add a discussion of this point.

We agree that T4P dynamics likely play a critical role in biofilm development in *A. baylyi* as they do in other species. However, we chose to focus on the role of T4P localization in biofilm formation for the purposes of this study, and it is the plan for future work to understand the regulation of dynamics and their role in this process. We have added discussion detailing this:

“While T4P dynamics are important for biofilm formation in many species, the factors that dictate T4P dynamics remain poorly understood. We thus used mutants lacking T4P dynamics in our studies to focus our efforts on understanding the function of T4P localization in *A. baylyi* biofilm development. Our findings demonstrate that T4P localization controls the spatial organization of multicellular communities by regulating the location and number of cell-cell interactions that occur. With this new understanding of how T4P localization influences the structure of microbial communities, future work can focus on identifying the factors involved in regulating T4P dynamics and the role of those dynamics in biofilm formation.” (lines 184-191).

Other points:

Line 45: It would be helpful to explain the functions of the genes involved in the Pil-Chp system in the Introduction.

We have added the putative functions of the genes in the Pil-Chp pathway to the introduction. We added:

“In *P. aeruginosa*, the methyl-accepting chemotaxis protein (MCP) homologue PilJ is thought to signal to the CheA homologue ChpA, and the CheW homologues PilI and ChpC are believed to act as intermediaries for Pil-Chp signaling leading to phosphoryl transfer from ChpA to the CheY homologues PilG and PilH. Other regulatory components encoded in the Pil-Chp operon include the putative transcriptional regulator ChpD as well as PilK and ChpB that are inferred to control the methylation state of PilJ.” (lines 54-59)

Line 56: The authors are surprised by the absence of an effect of pilJ deletion. Do they speculate that the localization does not respond to external stimuli? It would be good to discuss this point.

We have added discussion to address this point:

“In *A. baylyi*, FimV lacks the predicted periplasmic domain found in *P. aeruginosa*, in line with our results that the Pil-Chp pathway in *A. baylyi* does not regulate T4P synthesis. Instead, it is possible that truncation of FimV in *A. baylyi* influenced its evolution into a localization factor. While FimV is clearly a localization determinant of T4P machinery in *A. baylyi*, the details of FimV localization remain unclear. It is possible that filament-forming proteins including cytoskeletal elements may act as a scaffold for FimV localization, or it may be that that Pil-Chp signaling allows FimV to form a filament that localizes parallel to the long axis of the cell.

We show that deletion of the MCP homologue PilJ does not affect T4P localization. Canonical MCP proteins rest in the inner membrane and act as key signaling receptors during chemotaxis. Because PilJ does not play a role in Pil-Chp mediated T4P localization and no alternative MCP homologues are encoded in the genome, we hypothesize that an alternative, unidentified signaling protein fills this role (Figure 1b). Because PilJ is conserved in *A. baylyi* while other Pil-Chp genes have been lost (Extended Data 2), it is likely that PilJ acts as a signaling receptor for alternative Pil-Chp functions yet to be discovered.” (lines 170-183).

Line 110: The authors claim that *A. baylyi* is a nonmotile species. To my knowledge they are capable of twitching motility.

As mentioned above in our response to reviewer 1, we have not been able to detect twitching motility by *A. baylyi* in our lab conditions. We have added data to the supplement to demonstrate this point, and we have modified the text for clarity and to incorporate those data. Specifically, we added:

“In some species, including *P. aeruginosa*, twitching motility is thought to play a critical role in biofilm development. However, *A. baylyi* does not exhibit classical twitching motility under laboratory growth conditions (Extended data 7).” (lines 124-127).

Line 125: If the authors decide not to investigate the effect of pilus dynamics, then they should mention this in the abstract.

We have added this information to the abstract to highlight the fact that using mutants lacking T4P dynamics allowed us to specifically assess the role of localization in

multicellular development. We feel that the function of T4P dynamics and localization are two different biological questions and as such, we feel the use of $\Delta pilT$ mutants as a tool in this study is a strength and not a weakness because it allows us to separate those questions.

Specifically, we added to the abstract: “We further demonstrate through modeling and empirical approaches that subcellular T4P localization controls how individual cells interact with one another independently of T4P dynamics, with different patterns of localization giving rise to distinct multicellular architectures.” (lines 23-26).

Line 143: The discussion is overall very short. I suggest discussing the effect of subcellular type 4 pilus localization on multicellular interactions/colony development in comparison to other species with polar pili and peritrichous pili. Also, as mentioned before, the (putative) effects of pilus dynamics could be discussed at this point.

We have added a lengthy discussion including text focusing on the putative effects of T4P dynamics in biofilm formation by *A. baylyi* and the potential role of localization in other biofilm forming species:

“While T4P dynamics are important for biofilm formation in many species, the factors that dictate T4P dynamics remain poorly understood. We thus used mutants lacking T4P dynamics in our studies to focus our efforts on understanding the function of T4P localization in *A. baylyi* biofilm development. Our findings demonstrate that T4P localization controls the spatial organization of multicellular communities by regulating the location and number of cell-cell interactions that occur. With this new understanding of how T4P localization influences the structure of microbial communities, future work can focus on identifying the factors involved in regulating T4P dynamics and the role of those dynamics in biofilm formation. In many species including *P. aeruginosa*, which harbors polar T4P that are essential for biofilm formation, large three-dimensional biofilms are commonly observed. Our data show that T4P localization can influence multicellular architecture, and it is thus possible that the polar localization of T4P in *P. aeruginosa* plays a crucial role in its biofilm formation.” (lines 184-194).

Reviewer #3 (Remarks to the Author):

This work, titled “Subcellular localization of nanomachines regulates bacterial multicellular development” shows how the positioning of Type IV Pilus (Tfp) motors in a line along the cell major axis, is a driver of Acinobacter Baylyi bacterial cell organization within biofilms. The Pil-Chp chemosensory-like system is responsible for this particular Tfp subcellular localization by the aid of a protein, FimV, that connects Pil-Chp with Tfp. FimV is not part of the Pil operon as in *P. aeruginosa*. It is instead an orphan gene whose encoded product is here identified as a FimL interactant by pull-down experiments using FimL as a bait. Finally, a computational approach shows that this pili localization along the cell main axis is responsible of medium-size cell group formation, whereas a peritrichous distribution of pili would lead to the formation of larger groups. This work is novel and original and definitely adds insights on the current knowledge of Tfp biology and functions.

I believe that some additional data and information could improve the manuscript and clarify some aspects. Here are some proposed additions/changes.

We thank the reviewer for their thoughtful and directed suggestions and have incorporated changes throughout to address them. We feel this has greatly clarified and strengthened the manuscript. Please find details below addressing each comment:

MAJOR COMMENTS

The text can be highly improved by the additional of some information and clarifications.

Line 13: What are planar cell polarity proteins? Please briefly explain and add a reference.

We prefer to leave this introductory sentence without the specific protein details involved in planar cell polarity in mammals as we feel this information would distract from the introduction. However, we have added an additional reference that provides more information (Butler et al. Nature Reviews Molecular Cell Biology 2017).

Line 19: It took me a while to correlate the sentence “tfp appendages localize in a line along the long cell axis” with what I saw on Figure 1a. I first thought that the authors meant that the filaments (appendages) ran parallel to the long axis which is not what Fig 1a shows. This figure shows indeed that a maleimide stained duplicating cell (or a diplococcus?) have four foci running parallel to the main cell axis. Filaments extruding from these foci are perpendicular to this axis. Please, better describe this localization pattern in the abstract and throughout the text, specifying that motors (rather than the appendages) localize in a line along the cell axis. An illustration showing a typical *Acinetobacter baylyi* cell with the long cell axis indicated could also be added to Figure 1a.

We have changed the text to clarify this point. Specifically, we changed “we noticed that T4P localize in a line along the long axis of the cell body” to “we noticed that T4P are produced along a line that is parallel to the long axis of the cell body” (line 45).

We also added an extra figure to the supplement as the reviewer suggested for clarity (see new extended data figure 1).

Line 38: Why did the authors choose to use *Acinetobacter baylyi* as a model to study Tfp? What is known on the biology of this bacterium? What is its cell shape? Is it a pathogen? What is the current knowledge on their Tfp? Are these bacteria motile when they are not in a biofilm? If tfp have a role in adhesion beside in transformation, do they adhere to each other or to some cellular matrix?

We previously established *Acinetobacter baylyi* as a model organism for studying T4P (see Ellison et al. Nature Communications 2021). We have added more information about the biology of this species. Specifically, we have added “We recently developed the gram-negative coccobacillus-shaped bacterium *Acinetobacter baylyi* as a model for T4P study by introducing a cysteine residue into the major pilin ComP for subsequent labeling with fluorescent, thiol-reactive maleimide dyes. *A. baylyi* is a soil-dwelling species that is closely

related to emerging pathogens like *Acinetobacter baumannii*, both of which use their T4P to take up DNA from the environment to acquire antibiotic resistance genes.” (lines 39-44).

We likewise reference and have added new data (see new extended data 7) showing that *A. baylyi* are nonmotile in laboratory conditions including those used in this paper. Specifically, we added:

“In some species, including *P. aeruginosa*, twitching motility is thought to play a critical role in biofilm development. However, *A. baylyi* does not exhibit classical twitching motility under laboratory growth conditions (Extended data 7). To explore general principles of how the geometric distribution of T4P might affect multicellular interactions in nonmotile cells, we first used a computational approach.” (lines 124-128).

Additionally, little is known about the role of *A. baylyi* T4P in adhesion or the factors involved in biofilm formation in this species. Both of these topics are of interest for future study.

Figure 1d and 2b: It is useless to show 3 or 4 small cells when we have the histogram for statistical purposes. It is better to show one representative cell per strain where details are visible, especially for strains where pili are “dispersed”. Do pili from $\Delta pilT$ strains localize in one line (one dimension) along the long cell axis or, because cells of this strain are hyper-piliated, pili occupy a larger surface as compared to wt? Would electron microscopy unveil these differences?

We prefer to keep representative images of a few cells, but we have also added a “zoomed-in” version of representative cells of each strain to enhance the detail as suggested (see modified figure 1d). The $\Delta pilT$ hyperpiliation does not affect the dispersed phenotype as evidenced by the parent $\Delta pilT$ strain and double mutant strains containing Pil-Chp mutations that do not impact long-axis localization of T4P localization (the parent $\Delta pilT$, $\Delta pilH \Delta pilT$, or $\Delta pilJ \Delta pilT$).

Line 54: I understand the use of $\Delta pilT$ to better visualize pili. However, authors should show that pil-chp mutations have similar effects in wildtype as in $\Delta pilT$. It does not matter if differences are less pronounced in $\Delta pilT$ than in wt and if wt data are shown as supplementary information. On the same line, in Figure 4c and 4d, what is the effect of a pilT deletion on biofilm formation as compared to wildtype? Pilus dynamics might be disturbing for the computational analysis, yet they are physiological and must play an important role in biofilm formation. So, are differences between biofilms of pil-chp mutants and the parental strain still relevant in a wildtype background as in $\Delta pilT$?

We agree that T4P dynamics likely play a critical role in biofilm development in *A. baylyi*. However, we chose here to focus on the role of localization in biofilm development, and it is the plan for future work to understand the regulation of dynamics and their role in this process. We have added discussion detailing this:

“While T4P dynamics are important for biofilm formation in many species, the factors that dictate T4P dynamics remain poorly understood. We thus used mutants lacking T4P

dynamics in our studies to focus our efforts on understanding the function of T4P localization in *A. baylyi* biofilm development. Our findings demonstrate that T4P localization controls the spatial organization of multicellular communities by regulating the location and number of cell-cell interactions that occur. With this new understanding of how T4P localization influences the structure of microbial communities, future work can focus on identifying the factors involved in regulating T4P dynamics and the role of those dynamics in biofilm formation. In many species including *P. aeruginosa*, which harbors polar T4P that are essential for biofilm formation, large three-dimensional biofilms are commonly observed. Our data show that T4P localization can influence multicellular architecture, and it is thus possible that the polar localization of T4P in *P. aeruginosa* plays a crucial role in its biofilm formation.” (lines 184-194).

Line 56: Deletions in pilJ have no effects on pili localization. Are other MCPs in *Acinobacter baylyi* that could signal to the CheA-like? Otherwise, what is the hypothesized function for PilJ? Please discuss.

We have added discussion addressing this point. “We show that deletion of the MCP homologue PilJ does not affect T4P localization. Canonical MCP proteins rest in the inner membrane and act as key signaling receptors during chemotaxis. Because PilJ does not play a role in Pil-Chp mediated T4P localization and no alternative MCP homologues are encoded in the genome, we hypothesize that an alternative, unidentified signaling protein fills this role (Figure 1b). Because PilJ is conserved in *A. baylyi* while other Pil-Chp genes have been lost (Extended Data 2), it is likely that PilJ acts as a signaling receptor for alternative Pil-Chp functions yet to be discovered.” (lines 177-183).

Line 70: Are strains shown in 2b also in a Δ pilT background? If yes, please, specify.

The strains used in 2b are not Δ pilT mutants. We have added a line to the end of each figure legend indicating that detailed strain genotypes for strains used in each figure panel can be found in the strain table Table S1.

Lines 81-90. The pull-down is a huge experiment that allowed to uncover the *A. baylyi* homolog of FimV, which is a very important result. This experiment deserves more emphasis maybe by adding a supplementary table with the hits.

This experiment unfortunately did not pull down other significant hits likely affiliated with FimL, and thus we opted not to include a table containing only one protein hit. We interpret this to mean that pull-down conditions could likely be optimized and repeated for future study to find additional interacting partners involved in this pathway.

MINOR COMMENTS

Line 16: Beside “growth and motility”, add “cell shape and differentiation”.

We have included this addition.

Line 36: Please use the term peritrichous instead than dispersed here and throughout the text.

We prefer not to use the term peritrichous to refer to T4P here, as this term has only been used in the literature in reference to flagellar number and positioning (along with atrichous and monotrichous). Because T4P extension and retraction are dynamic processes that result in constant changes in the number of T4P fibers on the cell surface, we feel using this term may be confusing.

Line 51: Please specify either in the text or in the figure legend the correspondent flagellar chemotaxis homologs.

We have added this information to the text. Specifically, we added: “In *P. aeruginosa*, the methyl-accepting chemotaxis protein (MCP) homologue PilJ is thought to signal to the CheA homologue ChpA, and the CheW homologues PilI and ChpC are believed to act as intermediaries for Pil-Chp signaling leading to phosphoryl transfer from ChpA to the CheY homologues PilG and PilH. Other regulatory components encoded in the Pil-Chp operon include the putative transcriptional regulator ChpD as well as PilK and ChpB that are inferred to control the methylation state of PilJ.” (lines 54-59)

Legend to Figure 3a: specify that pili are maleimide labelled.

We have added this information to the legend.

Line 88: Any hypothesis on why FimV is truncated in *A. baylyi*? What is the fimV genetic environment?

We have added discussion about the possible role of FimV truncation in *A. baylyi*: “We show that subcellular localization of T4P in *A. baylyi* is dependent on a truncated homologue of the pilus-associated protein FimV found in *P. aeruginosa* in which it regulates T4P assembly. In *P. aeruginosa*, FimV is a large 919 amino acid protein that is predicted to localize to the inner membrane with the C-terminus residing in the cytoplasm and the N-terminus resting in the periplasm. The periplasmic domain possesses a peptidoglycan-binding motif that is predicted to bind peptidoglycan and aid assembly of the outer membrane secretin of the T4P assembly complex. In contrast, the C-terminal cytoplasmic domain is predicted to be involved in intracellular signaling. In *A. baylyi*, FimV lacks the predicted periplasmic domain found in *P. aeruginosa*, in line with our results that the Pil-Chp pathway in *A. baylyi* does not regulate T4P synthesis. Instead, it is possible that truncation of FimV in *A. baylyi* influenced its evolution into a localization factor. While FimV is clearly a localization determinant of T4P machinery in *A. baylyi*, the details of FimV localization remain unclear. It is possible that filament-forming proteins including cytoskeletal elements may act as a scaffold for FimV localization, or it may be that that Pil-Chp signaling allows FimV to form a filament that localizes parallel to the long axis of the cell.” (lines 163-176).

Line 93: “localize similarly” instead than “colocalize”.

We have made this change.

Extendend Data4: quantification of dispersed pili and transformation efficiency are more appropriate ways to assess protein functionality.

We have added quantification of the dispersed pili phenotype to this figure (please see new extended data 5).

Figure 3d: Is the arrow going from pilus machinery to FimV inverted?

We tried to use arrows to indicate protein recruitment rather than genetic regulation. We made the legend clearer in this regard by including “Gray arrow indicates the dependence of FimV linear localization on the Pil-Chp pathway while black arrows indicate the recruitment of FimL/PilG and T4P machinery proteins to FimV”.

Line 232: what the need of Fiji if protein fusion phenotypes have been quantified manually?

We regularly use Fiji/ImageJ for image visualization and manual counting of particles by taking advantage of the default cell counter plugin that is included with the Fiji/ImageJ download. We have added this information to the methods. Specifically, we added “Cell numbers, the percent of cells making pili, the percent of cells making dispersed pili, and fluorescent protein fusion phenotypes were quantified manually using the cell counter plugin in ImageJ” (lines 281-282).

Line 288: “1 ul” is irrelevant without the dilution factor.

We have fixed this to include the dilution. Specifically, we changed this line to “Then, 1 μ l of overnight cultures were subcultured into 3 ml of fresh LB medium and grown for an additional 16 hours” (line 349).

Line 302: >50 cells

We have made this change.

All figure legends should contain strain names and genotypes.

We have added a line to the end of each figure legend indicating that detailed strain genotypes for strains used in each figure panel can be found in the strain table Table S1.

A schematic representation of the hypothetical protein organization of *Acinobacter baylyi* pili would be helpful.

We have added a schematic representation of the pili to new extended data figure 1.

Reviewer #1 (Remarks to the Author):

The authors addressed all my suggestions regarding the previous manuscript. Publication of the work is recommended.

Reviewer #2 (Remarks to the Author):

My comments have been adequately addressed and I recommend to publish the revised version of the paper. Beautiful work!

Reviewer #3 (Remarks to the Author):

In this new version of manuscript "Subcellular localization of nanomachines regulates bacterial multicellular development", most of my comments have been addressed and I thank and felicitate the authors for that.

One of my major concerns regarded the use of $\Delta pilT$ as parental strain. In this new version, the authors explain that the use of $\Delta pilT$ allows the study of pilus positioning and functions in colony formation independently on pilus dynamics. While I fully understand the reasoning behind the use of $\Delta pilT$, it remains hard to believe that the authors never looked at simple mutant strains and compared their phenotypes with those of wildtype.

The authors should at least add a supplementary figure showing the $\Delta fimV$ colony phenotypes in a wildtype background and if cell groups are too dynamics to be analyzed and/or no differences are resolved between strains, the $\Delta pilT$ background can be introduced as a tool to get rid of pilus dynamics and compare simulations with experimental data. This addition is important as the description of the FimV (and thus the pilus positioning) roles on colony formation represents one of the two most important findings of this manuscript.

I would highly recommend to publish the article after this small addition.

Response to reviewers

Reviewer 3:

In this new version of manuscript “Subcellular localization of nanomachines regulates bacterial multicellular development”, most of my comments have been addressed and I thank and felicitate the authors for that.

One of my major concerns regarded the use of Δ pilT as parental strain. In this new version, the authors explain that the use of Δ pilT allows the study of pilus positioning and functions in colony formation independently on pilus dynamics. While I fully understand the reasoning behind the use of Δ pilT, it remains hard to believe that the authors never looked at simple mutant strains and compared their phenotypes with those of wildtype.

The authors should at least add a supplementary figure showing the Δ fimV colony phenotypes in a wildtype background and if cell groups are too dynamics to be analyzed and/or no differences are resolved between strains, the Δ pilT background can be introduced as a tool to get rid of pilus dynamics and compare simulations with experimental data. This addition is important as the description of the FimV (and thus the pilus positioning) roles on colony formation represents one of the two most important findings of this manuscript.

We thank the reviewer for their comments and feedback on our revised manuscript. The lead author who performed the experiments recently started a new position at a different institution and is currently unable to collect new data including colony phenotypes in wildtype and Δ fimV mutants. However, we have expanded our discussion to incorporate potential phenotypic consequences about how dynamics may influence colony morphologies: “Biofilms with cells capable of T4P dynamics may reveal more interesting changes in multicellular architecture over time. Because cells use their T4P to stick to each other, dynamic T4P may provide an avenue for cells to reorient themselves in multicellular communities over longer temporal scales and in response to different environmental conditions.” (Lines 200-204).